# Chronic Variable Stress and Cafeteria Diet Combination Exacerbate Microglia and *c-fos* Activation but Not Experimental Anxiety or Depression in a Menopause Model

**DOI:** 10.3390/ijms25031455

**Published:** 2024-01-25

**Authors:** Nelly Maritza Vega-Rivera, Erika Estrada-Camarena, Gabriel Azpilcueta-Morales, Nancy Cervantes-Anaya, Samuel Treviño, Enrique Becerril-Villanueva, Carolina López-Rubalcava

**Affiliations:** 1Laboratorio de Neuropsicofarmacología, Dirección de Neurociencias, Instituto Nacional de Psiquiatría “Ramón de la Fuente”, Mexico City 14370, Mexico; vegquim2909@gmail.com (N.M.V.-R.); sinhueazpilcueta@gmail.com (G.A.-M.); qfb.nancy.cervantes@hotmail.com (N.C.-A.); 2Facultad de Química, Benemérita Universidad de Puebla, Puebla 72570, Mexico; samuel_trevino@hotmail.com; 3Laboratorio de Psicoinmunología, Dirección de Neurociencias, Instituto Nacional de Psiquiatría “Ramón de la Fuente”, Mexico City 14370, Mexico; lusenbeve@yahoo.com; 4Departamento de Farmacobiología, Centro de Investigación y Estudios Avanzados del IPN, Mexico City 14330, Mexico; clopezr@cinvestav.mx

**Keywords:** chronic variable stress, menopause model, obesity, anxiety, Iba-positive cells, *c-fos*

## Abstract

The menopause transition is a vulnerable period for developing both psychiatric and metabolic disorders, and both can be enhanced by stressful events worsening their effects. The present study aimed to evaluate whether a cafeteria diet (CAF) combined with chronic variable stress (CVS) exacerbates anxious- or depressive-like behavior and neuronal activation, cell proliferation and survival, and microglia activation in middle-aged ovariectomized (OVX) rats. In addition, body weight, lipid profile, insulin resistance, and corticosterone as an index of metabolic changes or hypothalamus–pituitary–adrenal (HPA) axis activation, and the serum pro-inflammatory cytokines IL-6, IL-β, and TNFα were measured. A CAF diet increased body weight, lipid profile, and insulin resistance. CVS increased corticosterone and reduced HDL. A CAF produced anxiety-like behaviors, whereas CVS induced depressive-like behaviors. CVS increased serum TNFα independently of diet. A CAF and CVS separately enhanced the percentage of Iba-positive cells in the hippocampus; the combination of factors further increased Iba-positive cells in the ventral hippocampus. A CAF and CVS increased the *c-fos*-positive cells in the hippocampus; the combination of factors increased the number of positive cells expressing *c-fos* in the ventral hippocampus even more. The combination of a CAF and CVS generates a slight neuroinflammation process and neuronal activation in a hippocampal region-specific manner and differentially affects the behavior.

## 1. Introduction

The prevalence of obesity, diabetes, and psychiatric disorders, such as depression, is high; hence, the probability of their comorbidity is considered a serious health problem, and it is estimated to increase in the following years [1,2,3]. Indeed, some reports indicate that women are more vulnerable to developing both psychiatric and some metabolic disorders than men [2,4], particularly since there is already an increase in prevalence during the menopausal transition [5,6]. Menopause marks the end of reproduction, and the risk of cardiovascular disorders increases significantly after menopause due to physiological changes that affect the cardiovascular system [5].

Evidence indicates that the consumption of a low-quality diet such as a cafeteria diet (CAF; high fat and high sugar) increases the risk of developing obesity, diabetes, and metabolic alterations that can be associated with psychiatric disorders such as anxiety and depression [7,8,9]. In an animal model of menopause, a CAF induced high levels of anxiety-like behavior, insulin resistance, and an altered lipid profile [10]. However, the effect of a CAF on neurogenesis and neuroplasticity has been poorly explored in a menopause model.

Consistent findings in humans and rodents suggest that a high-calorie diet is associated with hippocampal-dependent memory deficit [11,12] and impaired hippocampal synaptic plasticity and neurogenesis [13,14,15,16], suggesting that the hippocampus is susceptible to dietary insults. In addition, the hippocampus, particularly its ventral region, is sensitive to stress challenges [17,18,19], whereas the dorsal hippocampus is widely involved in regulating learning and memory [18,20,21,22,23]. Thus, it is feasible to consider that a CAF may impact hippocampal function and neuroplasticity differentially.

Interestingly, both stress and diet produce systemic inflammation. From this point of view, metabolic disorders induced by a CAF are linked to systemic inflammation due to an exaggerated production of pro-inflammatory cytokines in multiple brain regions associated with cognitive functions, such as the hippocampus [11,15,24,25,26]. Other reports have shown that pro-inflammatory cytokines like IL-1α, IL-β, IL-6, and tumor necrosis factor α (TNFα) promote microglial activation [27].

In addition, experimental neuroinflammation models suggest that estrogens can modulate microglial activation in addition to stress and a CAF [28,29,30,31,32,33]. Some studies have suggested that a deficiency of estrogens produced by ovariectomy (OVX) induces microglial activation [34], contributing to the progression of neuronal damage in neurodegenerative disease [35,36,37,38]. Interestingly, several reports have shown that the reduction in ovarian hormones by OVX regulates adult hippocampal neurogenesis [39,40,41], decreases synaptic connectivity, reduces dendritic arborization [42,43,44], and exacerbates the depressive phenotype in animal models of acute and chronic stress [45,46]. 

Considering that the menopausal transition is a period vulnerable to developing both psychiatric and metabolic disorders, both can be enhanced by stressful events, worsening their effects. The aim of the present study was to evaluate whether a CAF combined with CVS exacerbates anxious- or depressive-like behavior, neuronal activation, affects cell proliferation and survival, as well as microglial activation in middle-aged ovariectomized (OVX) rats used as a model of menopause.

In addition, we evaluated body weight as an index of weight gain; lipid profile and insulin resistance as an index of metabolic changes; corticosterone as an index of HPA axis activation; and pro-inflammatory cytokines IL-6, IL-β, and TNFα as an index of systemic inflammation.

## 2. Results

### 2.1. Effect of a Cafeteria Diet and Chronic Variable Stress on Body Weight Gain, Lipid Profile, Glucose, and Insulin

As previously reported, rats fed a cafeteria diet gained more weight than rats eating chow (*p* = 0.001). The effect of the cafeteria diet persisted in the stressed group (*p* < 0.001), albeit this group showed less weight gain than the CAF-CTL group (*p* < 0.006) (Figure 1). A two-way ANOVA test yielded the following values: diet F_1,32_ = 29.84; *p* < 0.001; stress F_1,32_ = 11.11; *p* = 0.002; and the interaction diet X stress F_1,32_ = 2.34, ns.

The main effects on lipids were induced by a CAF, since this diet increased all lipids (cholesterol, TG, LDL, and FFA) in rats stressed and not stressed (*p* < 0.05; Table 1). Stress alone decreased the lipid profile significantly (cholesterol, TG, HDL) in chow-fed rats (*p* < 0.05). In contrast, the CAF increased the lipid profile in control and stressed rats compared to chow-fed rats (*p* < 0.05).

Regarding the TG content in different tissues, Table 1 shows that in the muscle, the man effect was associated with stress (F:_1,32_ = 10.72, *p* = 0.003); the stressed groups showed the lowest levels of TG. In contrast, the main effect in the liver was for the diet (F:_1,32_ = 4.40, *p* = 0.04); the CAF-stress group showed the highest TG values. In the heart, the CAF reduced the TG level content (diet F:_1,32_ = 25.99, *p* < 0.001), particularly in stressed rats, which showed the lowest TG levels (*p* < 0.05). In the adipose tissue, the TG content decreased in stressed rats, being the lowest in the CAF-stress group (*p* < 0.05). In contrast, the highest level was observed in control rats fed the CAF (*p* < 0.01).

Regarding glucose (Table 2), the area under the curve showed that the CAF significantly increased glucose levels in both stressed and not-stressed rats (diet F:_1,16_ = 24.61, *p* < 0.001). In addition, the area under the curve for insulin (Table 2) did not vary significantly among groups, albeit the highest values were observed in the CAF-CTL and chow-stress groups. HOMA-IR determinations showed that the CAF only increases insulin resistance in not-stressed rats (t = 1.96, df = 9, *p* = 0.04).

### 2.2. Effect of Cafeteria Diet and Chronic Variable Stress on the FST and EPM

The diet induced a trend to increase the immobility behavior (F_1,32_ = 3.482, *p* = 0.07) in the FST (Figure 1B) In contrast, chronic variable stress increased the immobility behavior in both the chow-stress and CAF-stress groups (F_1,32_ = 69.490, *p* < 0.001). No additional effect of stress and diet was observed on immobility behavior.

In contrast, diet (F_1,32_ = 6.03, *p* = 0.02) but not stress (F_1,32_ = 2.90, ns) increased anxiety-like behavior in the EPM by decreasing the percentage of time in open arms (Figure 1C; *p* = 0.01) and increasing the percentage in closed arms (Figure 1D; F_1,32_ = 5.80, *p* = 0.02). No significant changes in the number of total crosses (Figure 1E) related to diet (F_1,32_ = 3.76, ns) or stress (F_1,32_ = 0.12, ns) were observed. The two-way ANOVA test showed that the effect of diet depends on stress, yielding the following values for the interaction diet X stress in the open arms test (F:_1,32_ = 4.63, *p* = 0.03).

### 2.3. Effect of Cafeteria Diet and Chronic Variable Stress on Corticosterone and Pro-Inflammatory Cytokines

Corticosterone levels increased in response to the stress protocol (F_1,32_ = 63.63, *p* < 0.001) but not in response to diet (F_1,32_ = 0.37, ns). As shown in Figure 2, stress increased corticosterone independent of diet (*p* < 0.001) compared with the non-stressed groups.

A similar pattern was observed with TNFα (Figure 2A) since, in both the chow-stress and CAF-stress groups, an increase in this cytokine (F_1,17_ = 9.90, *p* = 0.006) was detected. No changes were detected by diet alone (F_1,17_ = 0.01, ns) or the interaction (F_1,17_ = 0.89, ns). Regarding IL-β (Figure 2B), no significant changes related to diet or stress were detected. Finally, for IL-6, diet induced the main changes F: _1,17_ = 5.21, *p* = 0.03. In stressed rats, the CAF reduced IL-6 levels (*p* = 0.01). 

### 2.4. Effects of Cafeteria Diet and Chronic Variable Stress on Microglial Activation in the Hippocampus

Figure 3, shows the effect of diet and stress on the percentage of Iba-positive cells with retracted morphology (panel A). As observed, diet (F:_1,10_ = 10.64, *p* = 0.009) and stress (F:_2,10_ = 4.21, *p* = 0.04) increased the percentage of retracted cells without significant interaction between factors. The highest value was observed in the CAF-stress group but did not reach statistical significance compared to the CAF-CTL group. Interestingly, when data were analyzed per ventral and dorsal hippocampus (Figure 3C), the effect of the diet was found to be dependent on stress exposure in the ventral hippocampus (F:_2,10_ = 6.59; *p* = 0.01) but not in the dorsal hippocampus (F:_2,10_ = 2.10, ns). Two-way ANOVA yielded the following values for the ventral hippocampus regarding diet (F:_1,10_ = 8.85; *p* = 0.01) and stress (F:_2,10_ = 89.38; *p* < 0.001), whereas for the dorsal hippocampus values regarding diet were (F:_1,10_ = 10.75; *p* = 0.008), and regarding stress (F:_2,10_ = 1.08, ns).

For the analysis of Iba-positive ramified cells, Figure 3B reveals that the diet (F:_1,10_ = 14.59, *p* = 0.003) exerts the main effect independent of stress (F:_2.10_ = 4.29, *p* = 0.04). As noted, diet decreases the percentage of ramified cells in the CAF-CTL and CAF-stress groups compared with the chow-CTL and chow-stress groups (*p* < 0.005). When data were separated by the ventral and dorsal hippocampus (Figure 3D), in the ventral hippocampus, the reduction in the percentage of ramified Iba-positive cells was induced by diet (F:_1,10_ = 20.89, *p* = 0.001) and stress (F:_2,10_ = 34.53; *p* < 0.001) without the interaction reaching statistical significance. In contrast, in the dorsal hippocampus, diet (F:_1,10_ = 11.34, *p* = 0.007) but not stress (F:_2,10_ = 0.94, ns) was the factor that promoted the reduction in ramified cells. In the dorsal hippocampus, the lowest percentage of ramified cells was detected in the CAF-stress group compared with the chow-stress (*p* < 0.005) and CAF-CTL (*p* < 0.005) groups.

### 2.5. Effect of Cafeteria Diet and Chronic Variable Stress on c-fos Expression in the Dorsal and Ventral Hippocampus

A general analysis showed that both diet (F:_1,11_ = 12.68, *p* = 0.005) and stress (F:_1,11_ = 46.74, *p* < 0.001) increased the number of immune-positive cells expressing *c-fos* (Figure 4A). The highest values of *c-fos* immunoreactive cells were observed in the CAF-stress group compared with the chow-stress (*p* = 0.005) and CAF-CTL (*p* = 0.005) groups.

A second data analysis was conducted to detect differences between the ventral and dorsal hippocampus in *c-fos* immunoreactivity. As seen in Figure 4B, in the dorsal hippocampus, diet promotes a trend to increase the number of *c-fos*-positive cells independent of stress (F:_1,11_ = 4.24; *p* = 0.06). In non-stressed rats fed with the cafeteria diet (CAF-CTL), the diet increased the number of *c-fos* expressions (*p* = 0.05). In contrast, stress significantly increased *c-fos* immunoreactivity in both the chow-stress and CAF-stress groups (F:_1,11_ = 6.24; *p* = 0.02) in the dorsal hippocampus. Regarding the ventral hippocampus (Figure 4B), the analysis showed that both diet (F:_1,11_ = 6.23; *p* = 0.03) and stress (F:_1,11_ = 9.18; *p* = 0.01) increased the number of immune-positive cells expressing *c-fos*. However, the interaction did not reach statistical significance (F:_1,11_ = 1.22, ns). The highest values of *c-fos* expression were detected in the group subjected to CAF-stress compared with the chow-stress and CAF-CTL (*p* < 0.05) groups.

### 2.6. Effect of Cafeteria Diet and Chronic Stress on Cell Proliferation, Survival, and Maturity in the Hippocampus of Old Ovariectomized Rats

Cell proliferation and the survival of newborn cells were observed using Ki67 and BrdU, respectively. The results showed (Table 3) the absolute number of Ki67-labeled cells in the gyrus dentate of animals exposed to the hyperlipidic diet and CVS did not reveal significant differences between the groups by either factor: diet (F:_1,12_ = 0.13; ns) or stress (F_1,12_ = 4.09; *p* = 0.06); however, their interaction yielded a significant value (F:_1,12_ = 6.09, *p* = 0.03). The chow-stress group showed the lowest value of Ki67-labeled cells (*p* < 0.05) compared to the chow-control group.

Regarding the evaluation of cell survival, the analysis of the total number of BrdU-labeled cells (Table 3) revealed that there were no significant differences by diet, F:_1,12_ = 1.28, ns; stress, F:_1,12_ = 1.08, ns; or their interaction, F:_1,12_ = 0.14, ns.

Finally, DCX-labeled neurons analysis (Table 3) indicates non-significant changes among groupsdue to diet (F:_1,12_ = 0.01; ns), stress (F:_1,12_ = 0.29, ns), or the interaction between the factors (F:_1,12_ = 0.9, ns).

## 3. Materials and Methods

### 3.1. Animals

Middle-aged female Wistar rats (mean age 12–14 months) supplied from the vivarium of the National Institute of Psychiatry “Ramón de la Fuente Muñiz” were housed in standard laboratory cages under a 12 h light/12 h dark cycle (starting at 2200 h) at a temperature of 23 ± 1 °C and with free access to food and water. All procedures were performed following the Mexican official norm for animal care and handling (NOM-062-ZOO-1999) and approved by the Institutional Ethics Committee of the National Institute of Psychiatry “Ramón de la Fuente Muñiz” (NC-17074.0).

### 3.2. Ovariectomy

To reduce estrogen levels, 12- to 14-month-old rats showing irregular cycles suggestive of being close to estropause [34,47] and with endocrine aging were bilaterally OVX under anesthesia (2,2,2-tribromoethanol; administered by intraperitoneal injection, i.p.) as previously described [48]. Muscles and skin were sutured, and a topical antiseptic was applied directly to the wound, followed by an IM injection of antibiotic at a dose of 15 mg/rat. Once the surgery was completed, the animals were returned to their home cage and remained there for a 3-week recovery period [10,34]. After, animals were randomly assigned to an experimental group.

### 3.3. Diets

Two types of diet were offered to the rats according to Estrada-Camarena et al. (2020) [10]: (1) chow diet (200 g of Ladiet^®^ 5001, Petfood, Mexico City, Mexico) and fresh water (2 L) equivalent to 15% protein, 50% carbohydrates, and fat 35%; and (2) cafeteria diet consisting of a combination of corn fritters topped with cheese (Cheetos^®^, Sabritas, Pepsico, Mexico City, Mexico, 100 g), chocolate milk (Svelty^®^ milk, Nestlé, Mexico City, Mexico, and Choco-milk^®^ 1 L, Mead Johnson Nutrition, Mexico City, Mexico) plus the chow diet (100 g) and fresh water (1 L). This diet was equivalent to 4.9% protein, 41.41% carbohydrates, and 53% fat [10]. The amount of food and water consumed by rats was evaluated every 24 h by weighing the food remaining in the feeders. Food and water were always calculated for 5 animals per cage.

### 3.4. Chronic Variable Stress (CVS) Protocol

Rats were housed in groups of five and randomly exposed for 3 consecutive weeks to the different stressors: white noise, continuous light, soiled cage, stroboscopic light, water deprivation, cool room, restricted movement, and water and food deprivation, according to the protocol used by Vega-Rivera et al. [45]. Regarding control groups, the rats were handled twice daily in a separate room maintained without stress but under similar handling and housing conditions as the experimental groups at an equivalent time [45]. 

### 3.5. Forced Swimming Test (FST)

The Porsolt test was used to evaluate the depressive-like behavior induced by the CAF, CVS, or the combination of stressors in ovariectomized rats. The FST was conducted by introducing rats in individual Plexiglass cylinders (46 cm in height and 20 cm in diameter) filled with 30 cm of water at 23 ± 2 °C [49]. Two swim sessions were conducted as follows: an initial 15 min pretest followed 24 h later by a 5 min test, which was videotaped for later scoring. The first session of the FST was used as part of the chronic variable stress protocol, and in non-stressed rats (not subjected to the chronic variable stress), this session (pre-test) was omitted. After each swimming session, rats were towel-dried, placed in heated cages for 30 min, and returned to their home cages. The immobility, considered a sign of depressive-like behavior, was scored during the test session and was defined as the minimal movements to keep the snout above the water in a period of 5 s during the 5 min test session [50].

### 3.6. Elevated Plus-Maze Test (EPM)

The elevated plus-maze test (EPM) analyzed anxiety-like behavior based on animals’ natural fear of heights and open spaces [51]. The EPM test consists of a plus-maze-shaped device elevated 50 cm above the floor. Each arm of the maze is 50 cm (length) × 10 cm (width). Two opposing arms have acrylic walls (40 cm high, closed arms), whereas the others lack walls (open arms). At the beginning of the test session (10 min), the rat was placed in the center of the maze facing a closed arm; an arm entry was considered once the rat placed all four paws on it. The parameters registered were (a) the cumulative time spent in the open arms, expressed as a percentage of time; (b) the cumulative time in the closed arms, expressed as a percentage of time; and (d) the total number of arm crosses. The percentage of time spent in the open arms is considered an index of the anxiety level [51]. The percentage of the time spent in the respective arms was calculated based on the cumulative time that rats spent in each arm in relation to the total test time [10,51].

### 3.7. Experimental Design

Figure 5 shows the experimental design followed. Briefly, three weeks after OVX, rats were divided into four groups: (a) chow diet—control (chow-non-stressed), (b) chow diet—stressed (chow-stressed), (c) cafeteria diet—control (CAF-non-stressed), (d) cafeteria diet—stressed (CAF-stressed). For 21 days, animals were subjected to a protocol of chronic variable stress or were only manipulated in the vivarium (Figure 5) and received a cafeteria or chow diet. To evaluate the weight, all rats were weighed at the beginning of the protocol and every third day until the end of the protocol.

On Day 21, all rats were tested in the EPM (10 min-session) followed by the FST (5 min-session). After the behavioral tests, rats of each group were returned to their home cage and continued with their diet. On the 22nd day, food was removed, and after 8 h of fasting, a glucose load of 1.75 g/kg was administered orally for the oral glucose tolerance test (OGTT). Blood was collected through tail puncture at 0, 30, 60, and 90 min to quantify glucose and insulin (n = 4 to 5 per group). After OGTT, all rats were euthanized by decapitation, and blood was collected from the trunk for lipid, proinflammatory cytokines, and corticosterone determinations; liver, muscle, heart, visceral, and fat pads were removed, frozen, and stored at −80 °C until analyses. Simultaneously, brains were removed and post-fixed in paraformaldehyde (PFA) for 24 h. Brains were kept in 30% sucrose in phosphate buffer until sectioned (Figure 5).

Behavioral tests were performed in 10 rats per group; glucose, insulin, and cytokines measures were obtained from 5 rats per group; lipids and corticosterone were obtained from 10 rats per group; and immunohistochemical analysis were obtained from 3 to 4 rats per group.

### 3.8. Immunostaining Procedures

Brains were cut into 40 µm coronal sections on a microtome (LEICA SM2010; Leica Biosystems, Inc., Deer Park, IL, USA) and stored at 4 °C in a cryoprotective solution, then processed for immunohistochemistry until required. Immunohistochemistry for Ki67 detection (1:500; Abcam, Waltham, MA, USA), BrdU (1:500; BD-Pharmigen^®^, Franklin Lakes, NJ, USA), DCX (1:250; Abcam, MA, USA), *c-fos* (*c-fos* 1:500; Santa Cruz Biotechnology, Inc., Dallas, TX, USA), or Iba-1 (1:1000; Wako Chemicals, Co., Richmond, VA, USA) was performed using the peroxidase method in series of every sixth section, as previously reported [41,45,46,52]. All Ki67-, BrdU-, DCX-, *c-fos*-, and Iba-labeled cells were counted throughout the rostro-caudal extension of the hippocampus, using a light microscope (Leica microsystems, Wetzlar, Germany, Germany). In the DG, the quantification of BrdU-, Ki67-, DCX-, *c-fos*, and Iba-labeled cells was limited to the granular cell layer (GCL) and sub-granular zone (SGZ). The latter region was defined as a band, limited by three nuclei down from the apparent border between the GCL and the hilar region (H). To obtain the estimated total number of Ki-67-, BrdU-, DCX-, *c-fos*, or Iba-labeled cells, the resulting number of positive cells for each marker was multiplied by six [41,52,53,54,55].

### 3.9. Zoometry

The rats’ weight was measured weekly using a digital balance at the beginning and end of experiments (Oxo Good Grips Scale, Oxo International, NY, USA). After euthanizing rats, the subcutaneous fat pad and visceral fat (abdominal and gonadal pad) were removed and weighed [10,56].

### 3.10. Biochemical Assays to Evaluate Glucose, Insulin, Lipids, Corticosterone, and Pro-Inflammatory Cytokines

Glucose, lipid, and lipoprotein serum concentrations were determined by commercial kits (Spinreact, Girona, Spain) with an automatic AutoKem II analyzer, according to Treviño et al. [56,57]. The free fatty acids (FFA) concentration was determined according to the method described by Brunk and Swanson [58].

Plasma insulin concentrations were determined by an ELISA immunoassay (Diagnostica Internacional SA de CV, Zapopan, JAL, Mexico), with the resulting antibody–antigen complex assessed at 415 nm in a Stat fax 2600 plate reader (WienerLab Group, Rosario, Argentina). Insulin concentrations were obtained from a standard curve ranging from 0 to 20 mU/mL. The total area under the glucose and insulin curve (AUC) was calculated using the trapezoidal method following the procedure described by Treviño et al. [56,59]. HOMA-IR percentage was calculated according to the mathematical models used by Trevino et al. [56,60]. Biochemical data were obtained from two sets of experiments that included all groups of rats (n = 5).

Corticosterone (CORT) was determined by an ELISA kit according to the manufacturer’s instructions (AssayDesigns, Ann Arbor, MI, USA) and quantified in an ELISA reader (US BioTek Laboratories, Shoreline, WA, USA) in serum samples obtained from the tested groups.

The serum concentration of IL-β, IL-6, and TNF-α was determined by ELISA; cytokines were quantified using reference standard curves generated with rat recombinant protein (IL-β Cat. 501-RL), (IL-6 Cat. 506-RL), and (TNFα Cat. 510-RT) that were incubated in a 96-well plate overnight at 4 °C with a capture antibody (AF-501-NA), (MAB506R), (MAB510R). Non-specific binding was blocked by incubating with 200 μL of 3% bovine serum albumin (BSA) in phosphate buffer saline (PBS); 100 μL of the serum (dilution 1:2) was added for 4 h at 37 °C and the sample washed four times with 500 μL PBST. The biotinylated antibodies (BAF501), (BAF506), and (BAF510) were used for the immunoreaction made with streptavidin-HRP (Cat DY998) and tetramethylbenzidine as a substrate (Cat. T0440-1L, Sigma Aldrich, St. Louis, MO, USA); after 10 min, the reaction was stopped with 50 μL of 2.5 M sulfuric acid. Absorbance was measured at 450 nm using a microplate reader (US BioTek Laboratories, Shoreline, WA, USA).

### 3.11. Statistical Analysis

Data are expressed as the mean ± error standard of the mean. Sigma plot 12.0 program (12.3 version, Systat Software, Inc., Palo Alto, CA, USA) was used for statistical analysis. Data were analyzed by two-way ANOVA, taking diet and stress as factors; when necessary, data were normalized before the statistical analysis. Post hoc analyses consisted of the Holm–Sidack test, and only values of *p* ≤ 0.05 were considered significant.

## 4. Discussion

The present study showed that in middle-aged ovariectomized rats, a CAF induces anxiety and CVS depressive-like behavior. Both stressors, the CAF and CVS, activate microglia and *c-fos*, particularly in the ventral hippocampus; the combination of factors produced the maximal effect. Only stress increased corticosterone; no changes in cell proliferation, survival, or maturity were observed after applying stress.

As previously reported [10], the CAF increases body weight, visceral fat, and lipid profile independent of stress exposure, with non-stressed rats being more affected. The present data reinforce the idea that this strategy is useful for inducing signs of obesity in middle-aged ovariectomized female rats [10]. An increase in glucose level, but not insulin, was observed in rats fed with the CAF independent of the stress condition. These data contrast with other studies performed with rats fed a CAF for 2 or 3 months, where increases in insulin and glucose were reported [61,62]. In the present study, a short-term CAF (one month) was used to induce obesity. In CAF-non-stressed rats, insulin resistance was observed, suggesting that alterations in metabolism, as a consequence of diet, were occurring. The apparent discrepancy related to insulin levels suggests that long-term protocols with the CAF are necessary to induce changes in insulin levels, reflecting a profound effect on glucose metabolism.

In the present study, chow-stressed rats showed low levels of lipids in plasma and different tissues compared to the chow-non-stressed group. The stress response includes activation of the hypothalamus–pituitary–adrenal axis, increasing the corticosterone release and, via the autonomic nervous system, promoting lipid mobilization that provides energy to the organism for a rapid response (fight or flight) to the possible threat (reviewed in Godoy et al. [63]). Our data agree with this notion, suggesting that ovariectomized middle-aged rats subjected to chronic stress express a metabolic response adequate to a stress challenge. However, the HDL was lower in chow-stressed rats, suggesting that, in stressed rats, a metabolic alteration could be starting. In line with this, high corticosterone levels were detected in the groups subjected to the CVS protocol, and data from the literature indicate that chronic stress reduces HDL [64].

Our data showed that the combination of factors, CVS plus CAF, did not increase the lipid profile in plasma or tissue any further, contrasting with previous reports that indicate that the combination of a high-fat diet or variable high-palatable food with CVS increases lipids in plasma and alters metabolism [64,65]. The apparent discrepancy in results could be related to the temporality in which diet and stressors were applied. In the present study, the CAF was presented simultaneously with CVS in contrast to other studies, where the diet was initiated several weeks before applying the stress schedule, suggesting that unhealthy diets could sensitize the body to trigger more severe metabolic alterations. In line with this proposal, CAF-stressed rats showed the highest hepatic TG accumulation, suggesting that, in this group, a metabolic alteration is present and could be detonating a cascade of events that could lead to the development of the lipid metabolic disorder [64].

Metabolic alterations and psychiatric disorders are frequently detected, mainly anxiety, in women with cardiovascular diseases, diabetes, and obesity [1,2,3]. In the present study, the CAF induced anxiety- but not depressive-like behaviors. The data on anxiety are in line with previous reports [10] but contrast with other reports revealing that an unhealthy diet induces depressive-like behavior [66,67,68]. Zeeni et al. [65] reported that a CAF, but not high fat or high carbohydrate, prevents the development of anhedonia (a depressive-like behavior sign) induced by CVS. Our results are in line with this, suggesting the high reinforcing value of the CAF in stressed rats. Another non-excluding explanation can be reached from studies where long-term diet protocols are used, and the development of a depressive-like behavior is observed [65,67,68]. In these protocols, two months after a high-fat diet combined with chronic stress increases anxiety and depression-like behavior [68].

In the present study, chow-fed rats subjected to CVS exhibited an increase in anxiety and depressive-like behavior independent of diet, showing higher corticosterone levels and decreased cell proliferation compared with the non-stressed rats. These findings contrast with other studies showing no changes in corticosterone levels in response to CVS [46,69]. This difference could rely on the aging of rats: middle-aged vs. young. In line with this, it has been reported that, compared with young rats, aged rats exhibit an elevation in basal levels of corticosterone and an impaired negative feedback process to return to basal levels [70]. Further, corticosterone increase in aged rats can induce a down-regulation of the hippocampal glucocorticoid receptors and receptor binding, favoring dysfunction of the negative feedback of the HPA axis [71]. Regarding cell survival, no changes were found in response to stress, which contrasts with other studies [46]. Noticeably, newborn cells in middle-aged rats are lower than those found in younger ovariectomized rats reported in our previous study [34]. It is possible that age, in combination with the endocrine condition, masked the effect of stress.

The percentage of Iba-positive cells increased, and the analysis of microglia morphology suggests activation in response to the CAF. In fact, retracted cells increase with the concomitant decrease in ramified cells, which was detected in the dentate gyrus of the hippocampus, suggesting an inflammatory condition [34]. The present result is in line with previous reports that showed that a mixed diet (high carbohydrate and high fat) increases neuroinflammation measured by cytokines and Iba-positive cells in the prefrontal cortex and hippocampus [57,72,73].

The CAF, but not CVS, increased Iba-positive cells in both dorsal and ventral regions; noticeably, only the ventral region was affected by the combination of the CAF and CVS. It has been proposed that the dorsal hippocampus is related to learning and memory processes, whereas the ventral hippocampus is more related to stress and anxiety regulation [17,18,74]. The present data suggest that the CAF is a stressor that contributes to activating both memory- and stress-related areas, favoring the development of anxiety. Supporting this, a similar pattern of increase in *c-fos* activation was observed. The CAF increased *c-fos* in the dorsal hippocampus, and maximal increase was observed in the CAF plus CVS in the ventral hippocampus. Interestingly, it has been suggested that *c-fos* could be considered a marker of the neuroinflammatory response [75]. In this regard, studies indicate that glial cells such as astrocytes, oligodendrocytes, and microglia express the *c-fos* proto-oncogene [75,76,77]. In addition, evidence has revealed that a systemic inflammatory stimulus, such as the administration of lipopolysaccharide, triggers an inflammatory response in the brain associated with microglial cell activation and expression of *c-fos* in these cells [77]. Considering these studies, our results suggest that the increased expression of *c-fos* could be associated with the activation of Iba-positive cells and the development of anxiety. Specific studies are necessary to confirm that an increase in *c-fos* expression corresponds to an increase in glial cells.

An unexpected result was that from the measured cytokines, only TNFα increased slightly in response to CVS, and no further increase was observed in response to the CAF or the CVS plus CAF combination. The data contrast with other reports indicating that peripheral pro-inflammatory cytokines increase in response to the CAF and CVS [67,68]. The discrepancies observed could be related to methodological differences. For example, long-term protocols of stress and diet were applied, i.e., two months versus one month, and the order of stimulus applied [68] also changed. Furthermore, other studies reported that a CAF diet increases cytokines in the peripheral nervous system and the brain [57,61]. A limitation of the present study is that brain cytokines were not measured. However, the increase in Iba-positive cells’ immunoreactivity suggests that a neuroinflammation process is present, and a long-term protocol using both factors (CVS and CAF) could also increase peripheral cytokines.

In addition, growing evidence demonstrates that the corticosterone released during stress periods can prime hippocampal microglia and facilitate inflammatory responses to a subsequent inflammatory challenge [78,79]. In line with this, we found that chronic stress increased corticosterone levels and provoked activation of microglia and *c-fos*, mainly in the ventral but not in the dorsal hippocampus. The CAF further increased this effect, thus, it is feasible that the CAF could act as a stressor potentiating the CVS effects on microglia. These findings are consistent with studies demonstrating that obesity is associated with chronic inflammation [57,61,72,73]. In fact, in a previous study, it was found that the increase in Iba was not further modified by either acute stress exposure or ovariectomy in middle-aged rats [34]. In the present study, the CAF induced obesity and microglial activation, which was further enhanced by CVS, suggesting that the combination of factors is enough to produce a vulnerability condition that may predispose to developing anxiety and depressive-like behaviors.

In conclusion, CVS combined with the CAF promotes *c-fos* and microglial activation in the ventral hippocampus, which could contribute to the development of experimental anxiety and depressive behaviors in an animal model of menopause.

## Figures and Tables

**Figure 1 ijms-25-01455-f001:**
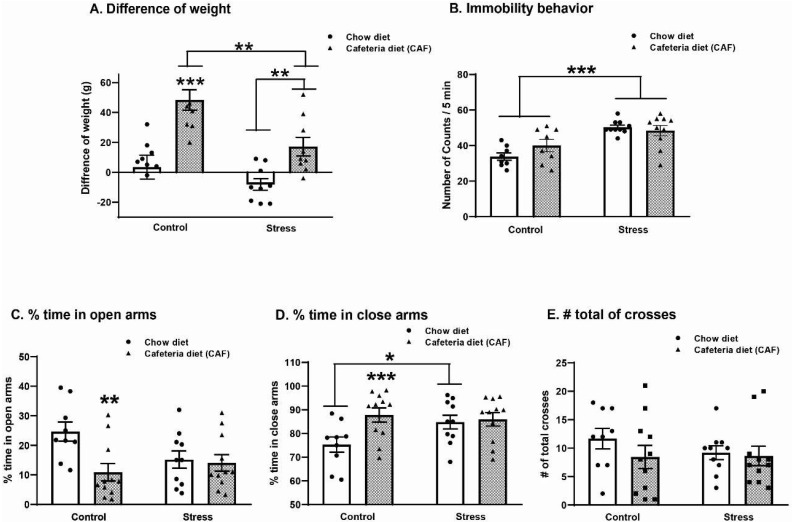
Effect of chronic variable stress (CVS) and cafeteria diet (CAF) in middle-aged ovariectomized rats on body weight gain (**A**), immobility behavior evaluated in the forced swimming test (FST; (**B**)), the percentage of time in open arms (**C**) and closed arms (**D**), and total number of crosses (**E**) evaluated in the elevated plus-maze (EPM). The clear bar represents the animals given a chow diet, and the dark bars represent the animals subjected to CAF. Data are expressed as the mean of the counts in a 5 min test period (FST) and the mean of percent of time (EPM) ± SEM. Data were analyzed with a two-way ANOVA and further application of the Holm–Sidak post hoc test, * *p* ≤ 0.05; ** *p* ≤ 0.01; *** *p* ≤ 0.001.

**Figure 2 ijms-25-01455-f002:**
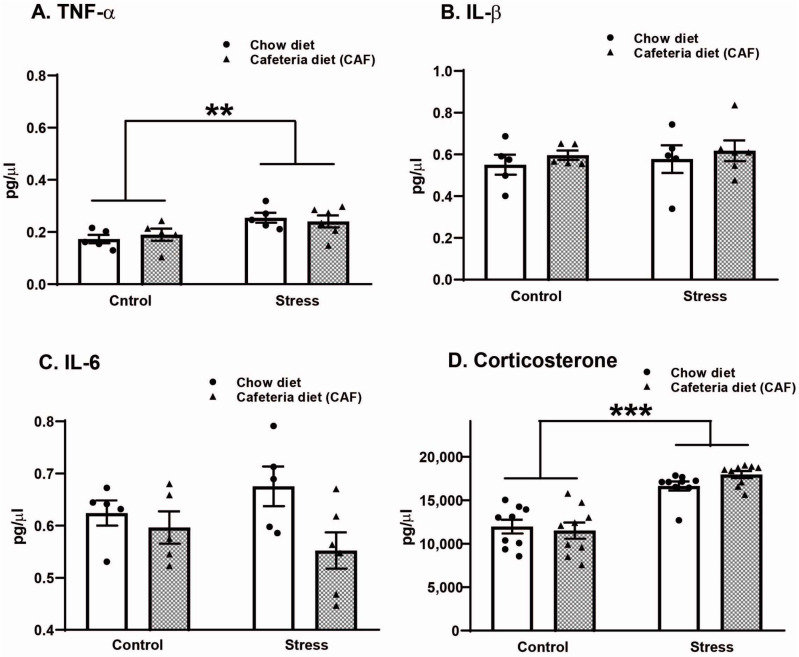
Effect of chronic variable stress (CVS) and cafeteria diet (CAF) on serum levels TNF-α (**A**), IL-1β (**B**), IL-6 (**C**), and corticosterone (**D**) in middle-aged ovariectomized rats. Data represent the mean ± SEM of n = 5–10 rats per group. Two-way ANOVA test followed by Holm–Sidack test, ** *p* < 0.01; *** *p* < 0.001.

**Figure 3 ijms-25-01455-f003:**
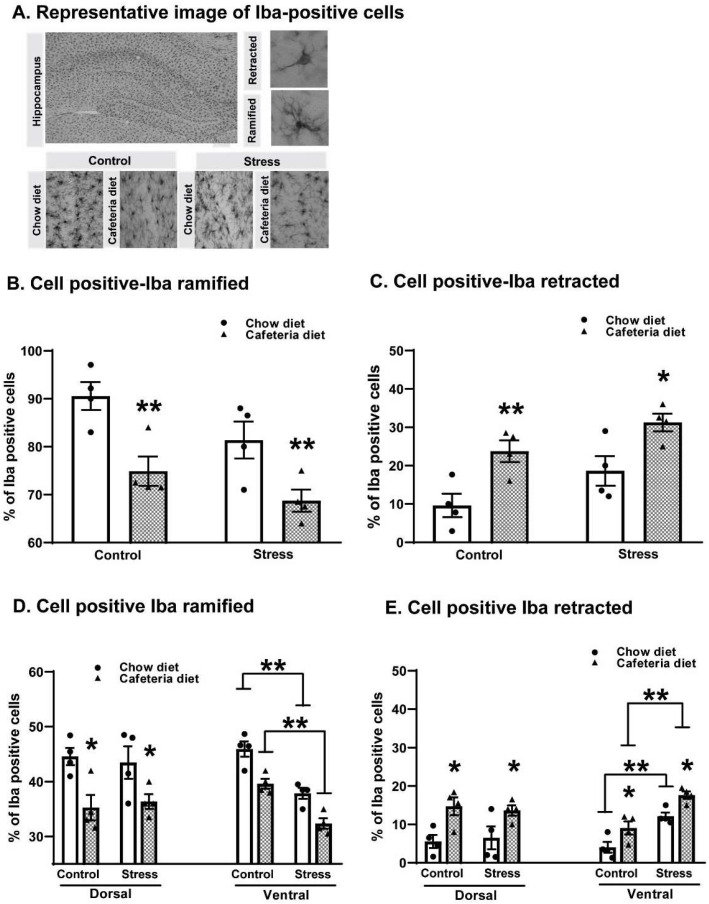
Effect of chronic variable stress (CVS) on microglial activation in middle-aged ovariectomized rats subjected to a cafeteria diet (CAF). (**A**) Representative photomicrographs of microglial cells distribution and their classification according to their morphology as ramified and retracted. Scale bars = 150 and 15 μm. Proportion of Iba-labeled cells with ramified (**B**) and retracted morphology (**C**) in the dorsal and ventral region of the dentate gyrus (DG) of the hippocampus (**D**,**E**). Data represent the mean percentage of Iba-positive cells ± SEM. Two-way ANOVA followed by Holm–Sidak, * *p* ≤ 0.05; ** *p* < 0.01.

**Figure 4 ijms-25-01455-f004:**
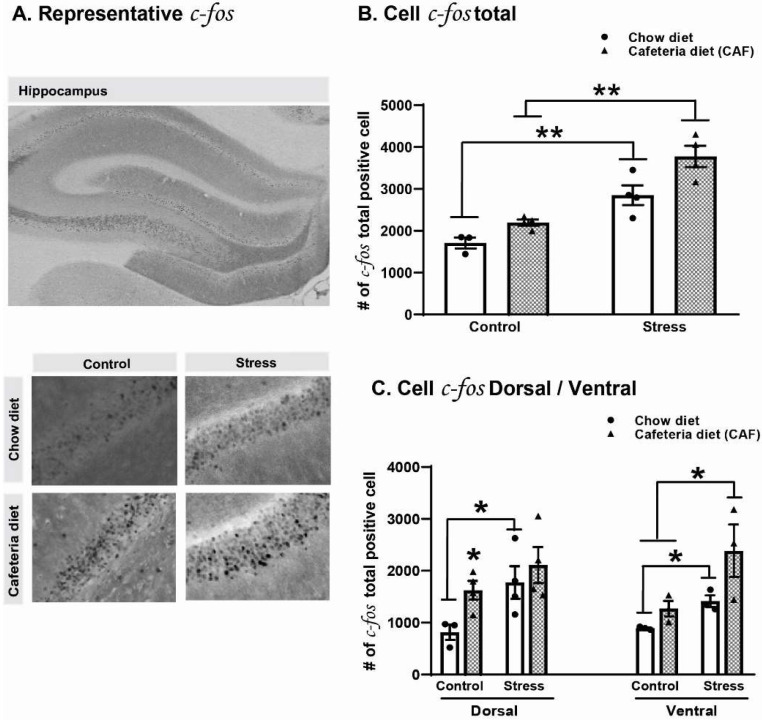
Effect of chronic variable stress (CVS) and cafeteria diet (CAF) on *c-fos* expression in the dentate gyrus (DG) of the hippocampus of middle-aged ovariectomized rats. (**A**) Representative images of *c-fos*-labeled cells distribution in the hippocampus’s dentate gyrus (DG). Scale bars = 150 and 15 μm. (**B**) Total number of *c-fos*-labeled cells in the DG (n = 4 per group). (**C**). Proportion of *c-fos*-labeled cells in the dorsal and ventral region of the DG of the hippocampus. Data represent the mean percentage of *c-fos*-positive cells ± SEM. Two-way ANOVA followed by Holm–Sidak, * *p* ≤ 0.05; ** *p* < 0.01.

**Figure 5 ijms-25-01455-f005:**
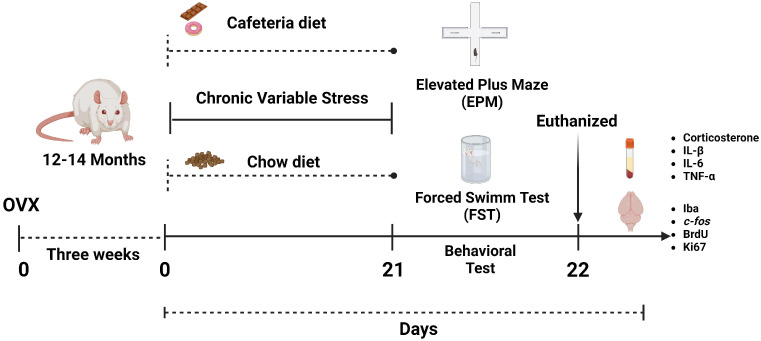
Schematic representation of the experimental design used to evaluate the effects of chronic variable stress (CVS) and the exposure to a cafeteria diet (CAF) on the anxiety- and depressive-like behaviors, neuronal activation, cell proliferation, survival, cytokines, and microglial activation in middle-aged ovariectomized (OVX) rats used as a model of menopause. Created in BioRender.com.

**Table 1 ijms-25-01455-t001:** Effect of cafeteria diet and chronic mild stress on lipid profile in plasma and tissue.

	Chow-CTL	Cafeteria-CTL	Chow-Stress	Cafeteria-Stress
**Lipid profile (mg/dL):**				
**Cholesterol**	94.11 ± 3.95	105.11 ± 3.23 *	71.00 ± 2.31 #	96.55 ± 5.63 **
**Triglycerides**	57.26 ± 6.55	72.14 ± 3.70 *	46.52 ± 6.88 #	72.00 ± 2.53 **
**LDL**	26.07 ± 1.96	38.90 ± 3.33 **	22.78 ± 2.16	31.80 ± 3.10 **
**HDL**	39.23 ± 3.10	48.55 ± 1.66 *	27.85 ± 1.59 #	44.61 ± 3.39 **
**vLDL**	28.16 ± 1.71	17.65 ± 2.04 **	20.35 ± 1.69 #	20.14 ± 1.08
**FFA**	26.93 ± 1.80	19.37 ± 1.60 *	26.37 ± 2.88	20.35 ± 1.73 *
**Tissue content of TG:**				
**Muscle**	16.25 ± 2.32	20.67 ± 1.26	12.84 ± 2.57	11.89 ± 0.46
**Heart**	22.08 ± 2.46	11.93 ± 1.16 *	18.55 ± 1.56	11.75 ± 1.09 *
**Liver**	21.06 ± 1.49	29.28 ± 2.60 *	22.57 ± 0.99	31.52 ± 7.55
**adipose**	41.92 ± 2.51	58.17 ± 4.78 **	31.21 ± 2.62	29.91 ± 1.42 #
**Visceral fat (mg)**	17.00 ± 1.95	35.22 ± 2.41 *	16.22 ± 3.08	30.55 ± 2.12 **
**Brown fat (mg)**	6.33 ± 1.15	11.22 ± 1.34 *	6.44 ± 0.68	12.33 ± 1.22 **

Data are presented as the mean ± SEM of 10 rats per group. Two-way ANOVA values followed by the Holm–Sidack post hoc test: * *p* < 0.05 Chow versus a cafeteria diet (CAF) in non-stressed rats; ** *p* < 0.01; # *p* < 0.05 CTL versus stress. Two-way ANOVA yielded the following values for **cholesterol** (diet F:_1,32_ = 23.32, *p* < 0.001; stress F:_1,32_ = 19.49, *p* < 0.001; interaction F:_1,32_ = 5.37, *p* = 0.02); for **triglycerides** (diet F:_1,32_ = 14.73 *p* < 0.001; stress F:_1,32_ = ns, interaction F:_1,32_ = ns); **low-density cholesterol** (LDL, diet F:_1,32_ = 15.32, *p* < 0.001; stress F:_1,32_ = 4.14, *p* = 0.05; interaction F:_1,32_, no significant); **high-density cholesterol** (HDL, diet F:_1,32_ = 25.75, *p* < 0.001; stress F:_1,32_ = 8.58, *p* = 0.01; interaction F:_1.32_ = ns); **very-low-density cholesterol** (vLDL diet F:_1,39_ = ns; stress F:_1,32_ = 10.7, *p* = 0.01; interaction F:_1.32_ = 9.48, *p* =0.01); **free fatty acid** (FFA, diet F:_1,32_ = 10.72, *p* < 0.001; stress F:_1,32_ = ns; interaction F:_1,32_ = ns). For the tissue content of triglycerides, the two-way ANOVA values were as follows for **muscle** (diet F:_1,32_ = ns; stress F:_1,32_ = 10.72, *p* = 0.003; interaction F:_1,32_ = ns); **for heart** (diet F:_1,32_ = 25.99, *p* < 0.001; stress F:_1,32_ = ns; interaction F:_1,32_ = ns); for **liver** (diet F:_1,39_ = 4.40, *p* = 0.04; stress F:_1,32_ = ns; interaction F:_1,32_ = ns); **for adipose tissue** (diet F:_1,32_ = 5.85, *p* = 0.02; stress F:_1,32_ = 39.78, *p* < 0.001; interaction F:_1,39_ = 8.07, *p* = 0.008).

**Table 2 ijms-25-01455-t002:** Effect of cafeteria diet and chronic mild stress on glucose and insulin profiles in plasma.

	**Chow-CTL**	**Cafeteria-CTL**	**Chow-Stress**	**Cafeteria-Stress**
**Glucose tolerance curve (mg/dL):**				
**0″**	122.40 ± 14.4	105.33 ± 5.68	110.16 ± 16.06	159.40 ± 11.11
**30″**	142.20 ± 7.6	240.60 ± 44.87	159.60 ± 4.57	215.00 ± 15.95
**60″**	134.80 ± 13.7	176.63 ± 12.18	155.00 ± 10.05	180.00 ± 22.37
**90″**	145.40 ± 9.6	143.60 ± 21.76	129.00 ± 6.53	193.00 ± 7.87
**AUC**	136.20 ± 18.08	165.84 ± 8.71 *	136.80 ± 8.53	186.85 ± 6.71 *
**Insulin curve (mg/dL):**				
**0″**	30.76 ± 3.9	54.15 ± 4.11	42.56 ± 10.22	26.30 ± 4.45
**30″**	59.96 ± 6.5	77.92 ± 9.06	72.96 ± 19.81	68.22 ± 2.39
**60″**	35.70 ± 10.5	54.35 ± 2.76	43.50 ± 15.27	63.26 ± 5.36
**90″**	62.88 ± 13.9	33.94 ± 8.49	36.00 ± 3.26	29.20 ± 3.79
**AUC**	47.37 ± 5.59	54.88 ± 4.53	51.43 ± 9.57	46.74 ± 2.40
**HOMA-IR**	1.57 ± 0.28	2.38 ± 0.28 *^1^	1.61 ± 0.19	1.69 ± 1.27

Data are presented as the mean ± SEM of 5 animals per group. Two-way ANOVA values followed by the Holm–Sidack post hoc test * *p* < 0.05 Chow versus cafeteria diet (CAF) in non-stressed rats; 0.05 CTL versus stress. ^1^ *t*-test *p* < 0.05. The two-way ANOVA values for the area under the glucose curve were as follows for diet (F:_1,16_ = 24.61, *p* < 0.001); and stress (F:_1,16_ = not significant; interaction F:_1,16_ = not significant); and the area under the insulin curve (F:_1.16_ = not significant; stress F:_1,16_ = not significant; interaction F:_1,16_ = not significant).

**Table 3 ijms-25-01455-t003:** Effect of diet and stress on cell proliferation, survival, and immature neurons of the dentate gyrus of the hippocampus in ovariectomized rats.

Group/Treatment	Cell Proliferation (+KI67)	Cell Survival (+BRDU)	Immature Neurons (+DCX)
Chow-control	255.0 ± 38.55	555.0 ± 56.5	1860 ± 245.5
Cafeteria-control	196.0 ± 19.29	595.7 ± 48.4	2319 ± 446.0
Chow-stress	161.5 ± 5.36 *	478.5 ± 59.7	2133 ± 336.7
Cafeteria-stress	205.2 ± 17.9	560.0 ± 50.1	1698 ± 211.3
Two-way ANOVA values	Diet F:_1,12_ = 0.13, ns Stress F:_1,12_ = 4.09, *p* = 0.06 Interaction F:_1,12_ = 6.09, *p* = 0.03	Diet F:_1,12_ = 1.28, ns Stress F:_1,12_ = 1.08, ns Interaction F:_1,12_ = 0.14, ns	Diet F:_1,12_ = 0.01, ns Stress F:_1,12_ = 0.29, ns Interaction F:_1,12_ = 0.9, ns

Data are presented as the mean ± SEM of four rats per group. The two-way ANOVA test was followed by the Holm–Sidack post-hoc test. * *p* < 0.05 versus chow-control group. BrdU = bromo-desoxiuridine; DCX = double cortin. ns = not significant.

## Data Availability

Data is available under specific request.

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
