# Peer review of "Chronic Variable Stress and Cafeteria Diet Combination Exacerbate Microglia and c-fos Activation but Not Experimental Anxiety or Depression in a Menopause Model"

_ijms, 2024, doi:10.3390/ijms25031455_

Round 1
Reviewer 1 Report
Comments and Suggestions for Authors
The authors used middle-aged ovariectomized (OVX) rats as a model of menopause to evaluate the effects of a combined cafeteria diet (CAF) and chronic variable stress (CVS). Although there have been various studies on the respective effects of CAF or CVS, the authors note in the introduction that there are few studies on the effects of CAF on neurogenesis and neuroplasticity in a menopausal model.
The authors sorted out the complexities of CAF and CVS, including the different phenomena that occur with CAF and CVS and the fact that the effects are offset when both stresses are added. 
The results obtained are complex but well organized.
With some minor modifications, I think this paper should be accepted.
For example, lines 414 and 417; what is HDL-C?
Misalignment of rows in Table 1
Author Response
The authors appreciate all reviewer’s comments and suggestions that undoubtedly improved the manuscript. As can be seen, the main changes were related to the careful review of the spelling, typos, and signs of the entire manuscript. We also checked the names and affiliations of the authors, acknowledgments, and funding data. In the next section, it is possible to find, point by point, the rebuttal letter.
The authors used middle-aged ovariectomized (OVX) rats as a model of menopause to evaluate the effects of a combined cafeteria diet (CAF) and chronic variable stress (CVS). Although there have been various studies on the respective effects of CAF or CVS, the authors note in the introduction that there are few studies on the effects of CAF on neurogenesis and neuroplasticity in a menopausal model.
The authors sorted out the complexities of CAF and CVS, including the different phenomena that occur with CAF and CVS, the effects are offset when both are added. 
The results obtained are complex but well organized.
With some minor modifications, I think this paper should be accepted.
For example, lines 414 and 417; what is HDL-C?
R= We apologize for the mistake; HDL replaced HDL-C and refers to the lipoprotein of cholesterol of high-density.
Misalignment of rows in Table 1
R=We appreciate the reviewer’s comment; Table 1 was fixed
Reviewer 2 Report
Comments and Suggestions for Authors
In this manuscript, Vega-Rivera and colleagues delineate the induction of anxiety by a cafeteria diet and depressive-like behavior by chronic variable stress. The authors illustrate that both the cafeteria diet and chronic variable stress activate microglia and c-fos, particularly in the ventral hippocampus, with the combination of factors producing the maximal effect. Moreover, they demonstrate that only stress leads to an increase in corticosterone, without observing changes in cell proliferation, survival, and maturity after applying stress. Overall, this is a highly innovative manuscript that pioneers the investigation into whether a cafeteria diet combined with chronic variable stress exacerbates anxious or depressive-like behavior, neuronal activation, affects cell proliferation and survival, as well as microglial activation in middle-aged ovariectomized rats used as a model of menopause—a topic that has been profoundly understudied and would be of significant interest to the readership of this publication.
This is a well-designed and well-written manuscript, providing novel insights that chronic variable stress combined with a cafeteria diet promotes c-fos and microglial activation in the ventral hippocampus, potentially contributing to the development of experimental anxiety and depressive behaviors in an animal model of menopause. Therefore, the findings presented in the current study have potentially high impact on the field.
However, I have one major concern that could enhance this manuscript: The authors state that behavioral tests were performed in 9 rats per group (line 165), but the n number in the behavioral test in Figure 2 is more than 9 in several groups. Could there be data inclusion/exclusion in the experiments? If so, please describe the inclusion/exclusion criteria in the methods.
There are several typos in the main text.
In line 253, ** p < 0.005 should be 0.01.
In line 309. **p < -0.01 should delete the hyphen.
Comments on the Quality of English LanguageThe entire manuscript needs to be carefully reviewed for grammar.
Author Response
The authors appreciate all reviewer’s comments and suggestions that undoubtedly improved the manuscript. As can be seen, the main changes were related to the careful review of the spelling, typos, and signs of the entire manuscript. We also checked the names and affiliations of the authors, acknowledgments, and funding data. In the next section, it is possible to find, point by point, the rebuttal letter.
In this manuscript, Vega-Rivera and colleagues delineate the induction of anxiety by a cafeteria diet and depressive-like behavior by chronic variable stress. The authors illustrate that both the cafeteria diet and chronic variable stress activate microglia and c-fos, particularly in the ventral hippocampus, with the combination of factors producing the maximal effect. Moreover, they demonstrate that only stress leads to an increase in corticosterone, without observing changes in cell proliferation, survival, and maturity after applying stress. Overall, this is a highly innovative manuscript that pioneers the investigation into whether a cafeteria diet combined with chronic variable stress exacerbates anxious or depressive-like behavior, neuronal activation, affects cell proliferation and survival, as well as microglial activation in middle-aged ovariectomized rats used as a model of menopause—a topic that has been profoundly understudied and would be of significant interest to the readership of this publication.
This is a well-designed and well-written manuscript, providing novel insights that chronic variable stress combined with a cafeteria diet promotes c-fos and microglial activation in the ventral hippocampus, potentially contributing to the development of experimental anxiety and depressive behaviors in an animal model of menopause. Therefore, the findings presented in the current study have potentially high impact on the field.
However, I have one major concern that could enhance this manuscript: The authors state that behavioral tests were performed in 9 rats per group (line 165), but the n number in the behavioral test in Figure 2 is more than 9 in several groups. Could there be data inclusion/exclusion in the experiments? If so, please describe the inclusion/exclusion criteria in the methods.
R= We appreciate the observation and the opportunity to clarify this point. Ten animals per group were used in the behavioral and lipid profile analysis. From these rats, 4-5 were also used for the glucose determination and the rest for the cytokine’s measures. This information was included in the methodology section, and the mistake related to the number of rats was corrected.
There are several typos in the main text.
R=The manuscript was carefully reviewed to detect and correct all typos.
In line 253, ** p < 0.005 should be 0.01.
R= the values were replaced.
In line 309. **p < -0.01 should delete the hyphen.
R=the hyphen was delayed.
The entire manuscript needs to be carefully reviewed for grammar.
R=the manuscript was carefully reviewed.